# User Manual for Coping Strategies Inventory Short Form (CSI-SF)—The Jackson Heart Study

**DOI:** 10.3390/ijerph21040443

**Published:** 2024-04-04

**Authors:** Clifton Addison, Brenda Jenkins, Monique White

**Affiliations:** Jackson Heart Study Graduate Training and Education Center, Department of Epidemiology and Biostatistics, School of Public Health, Jackson State University, Jackson, MI 39170, USA; brenda.w.campbell@jsums.edu (B.J.); monique.s.white@jsums.edu (M.W.)

**Keywords:** CSI-SF, validity, reliability, coping, Jackson Heart Study, African Americans

## Abstract

Researchers have asserted that patients who generally rely on powerful external sources to control their disorders can benefit from examining their coping mechanisms, which can potentially lead to a better understanding of the initiation and progression of some chronic diseases. By trusting their own internal powers and virtues, it is possible for some people to discover and navigate available strategies to balance and enhance their psycho-spiritual well-being and possibly their treatment and recovery. This review serves as a user manual for investigators who choose to use the CSI-SF to conduct their research on coping behaviors. The CSI-SF, which measures four coping strategies based on 16 items, was first assessed using the Jackson Heart Study (JHS) cohort in 2007. The reliability and construct validity of the CSI-SF was also later assessed among hemodialysis patients across 13 countries. In this study, the CSI-SF was assessed to be a reliable and valid instrument for measuring coping strategies. The CSI-SF serves the purpose of developing an inner voice that can assist with understanding how people cope with everyday life. The information gathered from the administration of the CSI-SF can inform investigators about environmental cues and triggers that can also impact individual health.

## 1. Introduction

Coping refers to a set of cognitive and behavioral strategies employed by individuals in response to stressors they encounter throughout their lives [1]. Many people experience events that are unpleasant or stressful; exposure to such events leaves them feeling overwhelmed, anxious, or worried due to the psychological burdens created by the challenging situations they encounter. Researchers have asserted that as patients rely on powerful external sources to control their disorders, in addition to their own internal powers and virtues [2], examining their coping mechanisms can lead to a better understanding of the initiation and progression of chronic diseases and uncover potential opportunities to balance and enhance their psycho-spiritual well-being and possibly their treatment and recovery. The Coping Strategies Inventory-Short Form (CSI-SF) was administered for the first time in the Jackson Heart Study Cohort during Exam 1, which spanned the period 2000–2004. The Jackson Heart Study (JHS), funded by the National Heart, Lung and Blood Institute (NHLBI) and the National Institute on Minority Health and Health Disparities (NIMHD) [3], was initiated as a longitudinal investigation of genetic and environmental risk factors associated with the disproportionate burden of cardiovascular diseases in African Americans. The JHS recruited 5306 African American residents living in the Jackson, Mississippi, metropolitan area of Hinds, Madison, and Rankin Counties. The JHS cohort was administered with the CSI-SF as one of the many instruments that measured psychological and behavioral risks associated with the high burden of cardiovascular disease in the African American community. Individuals’ responses to stressors covered by the CSI-SF included problem-related engagement and disengagement and emotion-focused engagement and disengagement. These types of coping strategies are credited with moderating the association between environmental risk factors and physical reactions that sometimes accelerate the occurrence of unhealthy outcomes, like chronic diseases.

## 2. Materials and Methods

The aim of this paper is to provide a high-quality user manual of the CSI-SF, which should be a critical component of survey instrument development and usage strategies. The CSI-SF user manual is advanced as a vital tool that could inform researchers about how to use the CSI-SF effectively to avoid potential errors and possible misuse in reporting results relating to the factor structure of the instrument. The importance of a user manual resides in the fact that it can save time for the researcher, thereby making it easier for researchers to understand the instrument, the constructs, the factor structure, and the scoring convention themselves so that they do not have to continuously rely on the authors or contact personnel to provide the details they need before they can initiate and finalize their own research prospectus. This manual is designed to support researchers with a useful guide that they can precisely follow every step of the way.

The CSI-SF, also called the Approach to Life B in the Jackson Heart Study, is a 16-item survey that was administered in the Jackson Heart Study (JHS) cohort of 5306 African Americans aged 35–84 years during the JHS Exam 1 data collection activities between 2000 and 2004. The 16-item CSI-SF was developed from the original 78-item Coping Strategies Inventory (CSI) and factor structure that was developed by Tobin et al. [4].

The CSI-SF aims to determine how people typically handle or cope with stress. The items in the CSI-SF represent thoughts or behaviors that people use to cope with stress. The responses to each item show how often a person copes with stress in the particular way mentioned. Some questions on the CSI-SF asked participants to respond about how much support they felt they received from family and friends. Other questions ask about how they deal with stress. Our investigators were interested in their honest answers to these questions because we wanted to know more about how they coped or dealt with stress, as research has shown that coping is associated with health in general, particularly heart disease [5,6].

In the instructions to participants, the JHS interviewers explained that “people often experience events that are unpleasant or stressful” and that the JHS was interested in understanding how they typically “handled or coped with stress”. In completing the CSI-SF, participants were asked to select their response from among the five Likert-scaled choices provided [7]. The interviewers read the questions to the participants, and the participants were encouraged to “answer what’s right for you” or “Answer what makes the most sense to you. The interviewers were trained to provide only neutral information regarding questions the participants had, like “use it the way you would normally use it in conversation”, or “Pick the answer that is closest to what is right for you” [7].

Most of the terms used in the various rating scales (e.g., Never to Almost Always) are left to the participants’ subjective perception of what these mean to them. They were asked to “use it the way that you would normally use it in conversation” or “pick the answer that is closest to what is right for you.” The participants were reminded that there were no right or wrong answers to the questions and that our investigators were simply interested in their views. Many of these questions were quite personal and could be perceived as sensitive or potentially embarrassing to the participant.

We knew that some individuals would have difficulty categorizing their responses for a variety of reasons, such as ‘it depends on the situation’. In such cases, the participant was encouraged to respond with regard to their “usual” behavior and which response was the most accurate for describing them across all types of situations. Responses were made based on a global frequency-based Likert-type scale ranging from Never (1) to Almost Always (5). The specific nature of the criteria that constituted a particular rating was determined subjectively by each participant.

### 2.1. Research Applications of CSI-SF

The CSI-SF measures four coping strategies based on 16 items: 4 items each indicating problem- vs. emotion-focused engagement or disengagement. Psychometric properties of the CSI-SF were first assessed using the JHS cohort in 2007 [8,9]. The reliability and construct validity of the CSI-SF was also later assessed among hemodialysis patients across 13 countries. For that study, the CSI-SF was translated into 9 languages and administered from 2009 to 2011 to >10,000 hemodialysis patients in 13 countries in phase 4 of the International Dialysis Outcomes and Practice Patterns Study (DOPPS)—a prospective cohort study of hemodialysis practices and outcomes that began in 1996. According to the investigators, the CSI-SF was chosen for the DOPSS for several reasons: (1) it includes only 16 questions, which is less than most of the other coping instruments, and it can be completed in only a few minutes; (2) the CSI-SF has been shown to be reliable and valid for measuring coping in persons with chronic disease; and (3) the conceptualization, structure, and content of the CSI-SF made sense to the investigators for its applicability to the general coping strategies that HD patients likely would utilize in living with end-stage kidney disease as a hemodialysis patient (i.e., it has strong content validity) [10].

The slightly modified English, German, and Swedish versions of the CSI-SF were assessed to be reliable and valid instruments for measuring coping strategies in hemodialysis patients. In addition, the CSI-SF Spanish version was also used as a test to provide a quick and efficient diagnosis of the coping strategies used in the face of stress in different settings [11].

### 2.2. The CSI-SF Instrument

The CSI-SF was developed as a brief 16-item scale derived from the original CSI [4,7,8]. The Coping Strategies Inventory (CSI) was originally constructed as a 78-item questionnaire [4]. The short form of the CSI was modified for use in the Jackson Heart Study as a 16-item version and was referred to as Approach to Life B. The CSI-SF was structured to reflect the original scale, with four 4-item subscales as follows: (a) Problem–Focused Engagement (items 1, 2, 8, and 9), (b) Problem-Focused Disengagement (items 4, 7, 12, and 14), (c) Emotion-Focused Engagement (items 5, 6, 11, and 13), and (d) Emotion-Focused Disengagement (items 3, 10, 15, and 16).

Participants were asked to rate the general frequency with which they utilize each listed coping strategy on a Likert-type scale and to respond in the following manner: 1 = “Never”, 2 = “Seldom”, 3 = “Sometimes”, 4 = “Often”, and 5 = “Almost Always”. Individuals received scores for each first-tier subscale (Engagement and Disengagement: range = 8–40) as well as for each of the four second-tier subscales (Problem-Focused Engagement, Problem-Focused Disengagement, Emotion-Focused Engagement, and Emotion-Focused Disengagement: range = 4–20). The CSI-SF uses a two-axis model to classify coping strategies (commitment and avoidance) and objective categories of coping (problem-focused and emotion-focused) [7]. Table 1 presents the dimensions and factor structure of the CSI-SF.

## 3. Results

### CSI-SF Reliability and Validity

Coping styles were measured by using the Coping Strategies Inventory Short Form (CSI-SF), a validated 16-item instrument used to measure engagement and disengagement coping styles. Engagement occurs when a person actively confronts a stressor (e.g., “I tackle the problem head on”). Disengagement occurs when a person avoids a stressor (e.g., “I try not to think about the problem”). Each item was evaluated by using a five-point Likert scale (1 = never, 2 = seldom, 3 = sometimes, 4 = often, and 5 = almost always). Scores within each eight-item sub-scale were summed (range: 8–40). Cronbach’s α was 0.59 for the disengagement scale and 0.70 for the engagement scale in the Jackson Heart Study cohort [8]. Two additional tables (Table 2 and Table 3), further describing the coping dimensions, are presented below. Table 2 presents details of the CSI-SF survey items.

For all items in Table 3, the answer choices were never (coded 1), seldom (coded 2), sometimes (coded 3), often (coded 4), and almost always (coded 5).

## 4. Conclusions

Like most survey manuals, the CSI-SF user manual was developed to provide instructions and guidelines on how to review responses to the instrument, how to understand the factor structure, how to score the items and the factors, and how to analyze the results in order to obtain a description of the coping strategies used by individuals, groups, and communities. The CSI-SF user manual gives the prospective researcher an overview of what they can do with the instrument and enables them to develop a good mental model of how they can quantify the factors to assess how individuals, groups, and communities cope with the stress they encounter in their daily lives.

Investigators who use the CSI-SF to assess the coping strategies of participants should interpret the findings with an understanding of its limitations [12]. Readers of this manual should use it as a guide to gain an understanding of human responses to environmental stimuli and provocations. The CSI-SF serves the purpose of developing an inner voice that can assist with understanding how people cope with everyday life. Everyone experiences conflicts with the two opposing sets of extreme influences: those emanating from the physical, environmental world around us and those sweltering within ourselves—what we see, what we hear, what we touch, and what we feel. Understanding the participants’ responses to the CSI-SF will help them, as well as public health and healthcare professionals, understand and learn how to monitor triggers and cues and, subsequently, balance responses to the complexities and vagaries of the environment that characterizes the community and the population of interest. The information provided can inform investigators about how environmental cues and triggers can also impact individual health. Learning to mediate stress factors through positive coping styles can lead to the development of chronic disease prevention strategies that can be used as effective chronic disease management tools [13].

For the JHS, it was important to understand coping behaviors as a social determinant of health because psychological factors have been implicated as a risk factor for the development of atherosclerosis and other cardiovascular disease outcomes, contributing to incidence and general mortality in populations [14,15].

In a study by Schneider et al. (2023), coping skills were found to be associated with a worse quality of life in patients at risk of or with diagnosed heart failure [16]. Utilizing the CSI-SF to gauge the coping skills of individuals and community members can help researchers better understand the healthy coping skills that enable some people to handle stress in a manner that does not diminish their functioning. Understanding the impact of coping skills can facilitate the implementation of adequate positive prevention and intervention strategies that can potentially reduce negativity as well as unhealthy outcomes.

## Figures and Tables

**Table 1 ijerph-21-00443-t001:** Dimensions and Factor Structure of the Coping Strategies Inventory Short Form (*CSI-SF*) Used in the Jackson Heart Study.

Scales/Subscales and Factors	Subscales/Factors	Tiers	Number of Items
Engagement Coping		1st Tier Subscale	8
	Problem-Focused Engagement (1) I make a plan of action and follow it. (2) I look for the silver lining or try to look on the bright side of things. (8) I tackle the problem head on. (9) I step back from the situation and try to put things into perspective.	2nd Tier Subscale	4
	Emotion-Focused Engagement (5) I try to let my emotions out. (6) I try to talk about it with a friend or family. (11) I let my feelings out to reduce the stress. (13) I ask a close friend or relative that I respect for help or advice.	2nd Tier Subscale	4
Disengagement Coping		1st Tier Subscale	8
	Problem-Focused Disengagement (4) I hope the problem will take care of itself. (7) I try to put the problem out of my mind. (12) I hope for a miracle. (14) I try not to think about the problem.	2nd Tier Subscale	4
	Emotion-Focused Disengagement (3) I try to spend time alone (10) I tend to blame myself (15) I tend to criticize myself. (16) I keep my thoughts and feelings to myself	2nd Tier Subscale	4

**Table 2 ijerph-21-00443-t002:** Details of the CSI-SF Survey Items.

Details of the CSI-SF	
Dimension	CSI-SF Survey Items
EFE	I try to let my emotions out.
	I try to talk about it with a friend or family.
	I let my feelings out to reduce the stress.
	I ask a close friend or relative that I respect for help or advice.
PFE	I make a plan of action and follow it.
	I look for the silver lining or try to look on the bright side of things.
	I tackle the problem head on.
	I step back from the situation and try to put things into perspective.
PFD	I hope the problem will take care of itself.
	I try to put the problem out of my mind.
	I hope for a miracle.
	I try not to think about the problem.
EFD	I try to spend time alone.
	I tend to blame myself.
	I tend to criticize myself.
	I keep my thoughts and feelings to myself.

EFE = Emotion-Focused Engagement. PFE = Problem-Focused Engagement. PFD = Problem-Focused Disengagement. EFD = Emotion-Focused Disengagement.

**Table 3 ijerph-21-00443-t003:** The Survey Items of the CSI-SF.

	Survey Items	Coping Dimension *
1	I make a plan of action and follow it.	2
2	I look for the silver lining or try to look on the bright side of things.	2
3	I try to spend time alone.	4
4	I hope the problem will take care of itself.	3
5	I try to let my emotions out.	1
6	I try to talk about it with a friend or family.	1
7	I try to put the problem out of my mind.	3
8	I tackle the problem head on.	2
9	I step back from the situation and try to put things into perspective.	2
10	I tend to blame myself.	4
11	I let my feelings out to reduce the stress.	1
12	I hope for a miracle.	3
13	I ask a close friend or relative that I respect for help or advice.	1
14	I try not to think about the problem.	3
15	I tend to criticize myself.	4
16	I keep my thoughts and feelings to myself.	4

* 1 = Problem-Focused Engagement—items 5, 6, 11, and 13 in the CSI-SF Inventory; 2 = Problem-Focused Disengagement—items 1, 2, 8, and 9 in the CSI-SF Inventory; 3 = Emotion-Focused Engagement—items 4, 7, 12, and 18 in the CSI-SF Inventory; 4 = Emotion-Focused Disengagement—items 3, 10, 15, and 16 in the CSI-SF Inventory.

## Data Availability

Data are available upon reasonable request.

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
