# Peer review of "User Manual for Coping Strategies Inventory Short Form (CSI-SF)—The Jackson Heart Study"

_ijerph, 2024, doi:10.3390/ijerph21040443_

Round 1

Reviewer 1 Report

Comments and Suggestions for Authors

Dear Author, please recheck the below items:

In the Introduction part:

1. What specific stressors do individuals typically encounter in their life experiences, prompting the need for coping strategies, as discussed in the introduction? 2. How do the coping mechanisms examined in this study contribute to our understanding of the initiation and progression of chronic diseases, as mentioned in the introduction? 3. Could the authors elaborate on the rationale behind administering the Coping Strategies Inventory-Short Form (CSI-SF) to the Jackson Heart Study Cohort, and how it fits into the broader objectives of the Jackson Heart Study, as described in the introduction?

In the method:

1. How were participants instructed to respond to the items on the Coping Strategies Inventory Short Form (CSI-SF), and what considerations were made regarding the subjective nature of the rating scales, as explained in the methods section?

2. Could the authors provide more insights into the process of assessing the reliability and construct validity of the CSI-SF within the Jackson Heart Study cohort and among hemodialysis patients in different countries, as mentioned in the methods?

3. How did the authors justify the selection of the CSI-SF for use in the International Dialysis Outcomes and Practice Patterns Study (DOPPS), and what specific characteristics of the CSI-SF contributed to its suitability for measuring coping strategies in hemodialysis patients, as discussed in the methods?

I did not see the Discussion part.

In the conclusion part:

1. How can the insights gained from participants' responses on the CSI-SF assist both individuals and healthcare professionals in understanding and managing triggers and cues that affect coping strategies, as discussed in the conclusion?

2. Could the authors elaborate on how the information provided in this manual could inform investigators about the impact of environmental cues and triggers on individual health, as mentioned in the conclusion?

3. What implications do the study findings and the use of the CSI-SF have for the development of chronic disease prevention strategies and their application as effective chronic disease management tools, as highlighted in the conclusion?

  1.  
  2.  

Author Response

Thank you very much for your input and your recommendations and suggestions. I have attached a document in which I have made revisions that I hope will be satisfactory to you.

Reviewer 2 Report

Comments and Suggestions for Authors

The article titled "User Manual for Coping Strategies Inventory Short Form (CSI-SF) - The Jackson Heart Study" focuses on the CSI-SF tool as a research method for coping behavior. It is presented as a manual for researchers interested in studying coping behaviors, based on the use of CSI-SF to measure four coping strategies through 16 items. The article details the development and validation of the instrument within the Jackson Heart Study (JHS) and among hemodialysis patients from 13 countries, highlighting the reliability and validity of CSI-SF in measuring coping strategies.

Introduction

It is well-structured and effective in establishing the context and importance of the coping strategies study. I appreciate the way the concept of coping is approached, highlighting its relevance in understanding the progression of chronic diseases and psycho-spiritual well-being.

A strong aspect of the introduction is the detailed presentation of the background of the Jackson Heart Study (JHS), emphasizing its purpose and importance in heart health research among African Americans. This provides a solid foundation for understanding the reasons why CSI-SF was chosen as an instrument in this study, enhancing the credibility and relevance of the research.

There are a few areas that could be improved to increase clarity and impact:

- The introduction could benefit from a more detailed discussion about previous works in the field of coping strategies, highlighting the specific gaps this study aims to address (this in the part of the introduction or the creation of a separate chapter for the review of the specialized literature).

- Although the importance of the coping study in the context of chronic diseases is mentioned, a more detailed argument linking the objectives of CSI-SF directly to the needs identified in literature and society could be useful.

- Although the general context is established, it could be clarified by explicitly listing the main objectives of the current study to distinguish them more clearly from the objectives of the JHS.

Materials and Methods

The chapter provides essential details about the research instrument and methodological approach, but there are several areas where it could be improved to increase clarity and scientific rigor:

- The description of the studied population and the inclusion or exclusion criteria could be more detailed. This would help understand the context of the population and the applicability of the results.

- Additional details about the data collection process, such as the duration of administering CSI-SF or the way responses were collected, could improve methodological transparency.

- The chapter would benefit from a more detailed section about the statistical methods used for data analysis. Mentioning specific tests, validity and reliability criteria, and how missing data were treated would add to the study's rigor.

Results

The chapter presents the research findings, including the reliability and validity of CSI-SF. This is a key element of any scientific study, as it provides empirical evidence that supports or contests the research hypotheses. Observing that the study does not include a section dedicated to discussions, considering these aspects, I believe this chapter should be improved, as follows:

- Include more statistical details, such as the exact values of the tests and confidence intervals, to increase transparency and allow a deeper evaluation of the results' robustness.

- Integrate a section that discusses the significance of the results in the context of the research hypotheses and previous studies, providing a perspective on the study's contribution to existing knowledge.

- Identify and discuss any unexpected results, exploring possible explanations and implications.

- Compare the results with those of other relevant studies, highlighting similarities and differences, to emphasize this study's unique contribution.

- Analyze and report any differences in results among the studied population's subgroups, to evaluate the applicability and universality of CSI-SF.

Conclusion

- The conclusion could benefit from a closer link with current discussions in literature and previous studies, to place the results in a broader academic context.

- Although implications are mentioned, the chapter could explore in detail how the results could be applied in clinical practice or in designing interventions.

- Future directions are mentioned, but these could be expanded to provide clearer guidance for subsequent studies, including possible areas of interest and unexplored research questions.

Additionally, any study presents research limitations. In this case, I recommend creating a separate chapter for these limitations.

Author Response

Thank you for your suggestions, recommendations, and input. I have made revisions that I hope will behave addressed your concerns.

Thank you. 

Round 2

Reviewer 2 Report

Comments and Suggestions for Authors

This review is in order. I am glad that my observations were taken into consideration. The report can be published in this form. I apologize for the oversight (I initially did not notice that it was a short report). I wish you all the best.